# Consumer and provider perceptions of the specialist unit model of care: A qualitative study

Janet C. Long[1]*, Ann Carrigan[1], Natalie Roberts[1], Robyn Clay-Williams[1], Peter D. Hibbert[1,2], Yvonne Zurynski[1], Katherine Maka[3], Graeme Loy[3], Jeffrey Braithwaite[1]

1 Australian Institute of Health Innovation, Macquarie University, Sydney, Australia, 2 IIMPACT in Health, Allied Health and Human Performance, University of South Australia, Adelaide, South Australia, Australia, 3 Western Sydney Local Health District, Sydney, New South Wales, Australia

* janet.long@mq.edu.au

## Abstract

### Background

Specialist care units cater to targeted cohorts of patients, applying evidence-based practice to people with a specific condition (e.g., dementia) or meeting other specific criteria (e.g., children). This paper aimed to collate perceptions of local consumers and health providers around specialist care units, as a model of care that may be considered for a new local healthcare facility.

### Methods

This was a qualitative study using two-hour workshops and interviews to collect data. Participants were consumers and health providers in the planned facility's catchment: 49 suburbs in metropolitan Australia. Consumers and health providers were recruited through advertisements and emails. An initial survey collected demographic details. Consumers and health providers participated in separate two-hour workshops in which a scenario around the specialist unit model was presented and discussion on benefits, barriers and enablers of the model was led by researchers. Detailed notes were taken for analysis.

### Results

Five consumer workshops (n = 22 participants) and five health provider workshops (n = 42) were conducted. Participants were representative of this culturally diverse region. Factors identified by participants as relevant to the specialist unit model of care included: accessibility; a perceived narrow scope of practice; coordination with other services; resources and infrastructure; and awareness and expectations of the units. Some factors identified as risks or barriers when absent were identified as strengths and enablers when present by both groups of participants.

**Data Availability Statement:** Raw data cannot be shared due to ethics stipulations that the site and individuals not be identified. The data set is restricted by the Macquarie University Human

Research Ethics Committee (2021/ PID01000) and can be requested at this email address: ethics.secretariat@mq.edu.au.

**Funding:** This project was funded by Health Infrastructure NSW under the grant HI20314. The funders had no role in the conduct of this study or its results, interpretation or presentation.

**Competing interests:** The authors have declared that no competing interests exist.

**Abbreviations:** LHD, Local Health District.

## Conclusions

Positive views of the model centred on the higher perceived quality of care received in the units. Negative views centred on a perceived narrow scope of care and lack of flexibility. Consumers hinted, and providers stated explicitly, that the model needed to be complemented by an integrated model of care model to enable continuity of care and easy transfer of patients into and out of the specialist unit.

## Background

The way that health care is delivered using different models of care has evolved over time. Traditional models of care have changed and adapted, for example, as bed space becomes limited [1, 2], new technology is introduced (e.g., minimally invasive 'keyhole' laparoscopic surgery replacing more invasive laparotomy methods [3]) and increasing knowledge of best practice is gained. Innovative models of care such as use of remote web-based monitoring or hospital in the home style services are offering alternatives to usual care delivery. There is growing evidence of the advantages of innovative models of care over usual care, positively affecting clinical outcomes [e.g., 4, 5] quality of life outcomes [6], and decreasing readmission rates [7, 8].

One of these innovative models of care is the specialist hospital or unit. Specialist hospitals or care units cater to targeted cohorts of patients, diagnosing and managing people with a specific condition or meeting other specific criteria (e.g., children). This is in contrast to more traditional models where patients with a much broader range of conditions were cared for in the same unit (e.g., surgical, medical). A key difference of specialist care units with other models of care such as integrated care, is that all the health professionals sharing care of the patient are co-located. The philosophy behind specialist units is that they use interdisciplinary teams to provide expert, best practice for that condition, and these teams are co-located in the one unit–either inpatient ward or outpatient (ambulatory) clinic. In Australia, specialist units are a key part of the publicly funded health system, strategically incorporated into hospitals and community-based facilities based on a state by state assessment of need, prevalence and access to resources. Examples are: specialist children's hospitals (centralised in capital cities), specialist inpatient units for people with an acute spinal cord injury [9], or multidisciplinary specialist services for people with end stage renal disease [10].

Empirical data supports the effectiveness of this model. Two separate reviews of the specialist unit model of care for people with chronic kidney disease found improved estimated glomerular filtration rate, lower hospitalisation rates for end stage kidney disease and reductions in overall mortality compared to a matched cohort who received usual care [11, 12]. Two other reviews on the same model but for patients with heart failure, gave evidence for reduced rates of all-cause readmissions [13, 14], and heart failure-specific readmissions [13]. In all cases, superior outcomes were attributed to expert, interdisciplinary collaboration allowing for comprehensive assessment, treatment and timely follow-up.

### Context

A planned new public health facility to be built in a metropolitan site in Australia provided an opportunity to consult widely with local community members and health providers around different models of care the facility could consider incorporating. The consultation involved presenting different, innovative models of health care and eliciting feedback from consumers

and health providers separately via two hour workshops. Models of care and any evidence of their impact on outcomes were first compiled and reported through a review of the peer-reviewed and grey literature [15]. Models of care identified were: ambulatory care, digital hospital, hospital in the home, integrated care, virtual care, consumer focussed care, and the specialist hospital or care unit. This paper focusses on the consultation involving the specialist unit model of care.

While literature on the clinical outcomes and costs of innovative models of care was relatively easy to find, consumer perceptions of the usefulness or relative advantage of the models was not. Health service providers' perceptions of working within these models is also an under-researched area. Understanding community views of different models of care–especially the barriers to accessing care–is an important part of the design of new facilities. Being able to consult on a planned facility before it had been designed gave the research team a unique opportunity to explore this under-researched area and feed into future plans.

The aim of this study was to report on local consumer and health provider perceptions of specialist care units as a model of care. Such an option is being considered for a new healthcare facility to be built in a metropolitan setting in New South Wales, Australia.

## Methods

We undertook a qualitative study of consumer and provider views and preferences in relation to the specialist unit model of care delivery for a new public health facility. Study methods for the entire consultation procedure are described in detail elsewhere [16] while specific methods for this part of the study are given in detail below. All methods were carried out in accordance with relevant guidelines and regulations. COREQ Checklist is provided in S1 File. Ethics approval for conducting the study was obtained from the Local Health District Human Research Ethics Committee (2021/ PID01000).

### Procedures

**Recruitment.** Recruitment and workshops occurred between July and October of 2021. Local community members were recruited for the consumer workshops through advertisements in local papers, Facebook invitations and emails circulated to people connected to the Local Health District (LHD). Eligible people lived within the new facility's proposed catchment (49 suburbs) in Sydney, New South Wales, Australia, as defined by the LHD's planning team on 16th July 2021. Health providers were recruited via broadcast emails sent by the LHD.

Study advertisements contained a link to an online questionnaire hosted on the REDCap platform [17]. In accordance with the Australian National Statement on Ethical Conduct in Human Research [18], clicking on the link and completing the questionnaire was taken as implied consent for their data to be collected. The questionnaire was used to lodge participants' expression of interest in the study and asked for demographic data including age, gender, location, ethnicity, and contact information. Health providers were asked to specify their role and specialty, while consumers were asked about their health, including whether they had a chronic condition. All data was securely stored on the university's servers and was used only to report the demographics of participants collectively. No remuneration was offered for participation in the workshops or supplementary interviews. Everyone who submitted an expression of interest questionnaire and was sent written information about the study. All participants of the workshops gave formally written consent to take part.

**Workshops.** The specialist unit model was presented at five consumer and five health provider two-hour workshops. All these workshops were held in English. The workshops (held separately for consumers and health professionals) were held online using the Zoom

videoconferencing platform and were offered both within and outside of working hours. Planned face-to-face focus groups were not possible due to COVID lockdowns. Supplementary interviews with interested participants identified in the workshops were conducted online and over the phone.

Workshops were designed to elicit the strengths, barriers and enablers of the specialist care unit model. Two researchers (from a team of six) facilitated the workshops. One researcher introduced the topic and asked the questions while the other took detailed notes. To minimise bias, the researchers chosen for each workshop / model of care were different, but at least one of the pair of researchers for each group was a postdoctoral health service researcher, experienced in qualitative methods. After each workshop the detailed notes were reviewed and discussed by both researchers to agree they represented an accurate record of the proceedings.

Each workshop presented a scenario that illustrated the model of care, featuring a person with a high impact, high prevalence condition (e.g., dementia). After introducing the workshop, facilitators described the specialist unit model, alongside the scenario that presented the model of care in a patient context. Specialist dementia units were understood as being located within a larger public hospital.

> *Harrold is an 82-year-old man with mild dementia, who develops a urinary tract infection. He has been referred to a* **specialist dementia unit** *in a geriatric care ward at the local hospital. Harrold and his family are reassured that he will receive the highest level of evidence-based care for dementia from a specialised team of health professionals.*

After presentation of the scenario, the researchers asked questions from the perspective of the patient, then more generally, about the strengths and weaknesses of the model, whether it seemed appropriate and usable, and to identify any safety issues they might envision. Table 1 shows the wording of the questions. Health providers were asked the same questions but from

**Table 1. Schedule of questions used in the workshops after presentation of the patient scenario.**

| Consumer schedule of questions | Provider schedule of questions |
|---|---|
| We would like to ask questions from Harrold's as well as your own perspective. Let's start with Harrold: From Harrold's perspective:<br>1. What is good about this model for Harrold?<br>2. What about this model might make it difficult for Harrold? *Can you think of anything about it that might be impractical*? *Can you think of anything about it that might be unachievable*?<br>3. What needs to be in place for this to work for Harrold? *For example, systems, processes, people, skills and equipment*? Now from your perspective:<br>4. What about this model might be good for you and your family? Can you think of anything about it that might be impractical?<br>Can you think of anything about it that might be unachievable?<br>5. What about this model might make it difficult for you and your family?<br>6. How easy is this to use for you?<br>7. What would stop you using it?<br>8. Can you think of other people who would have difficulty using this model?<br>9. We have already asked for Harrold but what other things needs to be in place for this to work for you? For example, systems, processes, people, skills and equipment | 1. In an ideal world, how would Harrold's care be delivered? How could you best model this?<br>From your perspective:<br>How would this model help to solve the big problems for you? (What are the pros/strengths for you?)<br>What barriers limit this model for you?<br>What enablers would need to be in place for this to work?<br>From your patients' perspective:<br>How would this model help to solve the big problems for your patients?<br>What might be the pros/strengths?<br>What barriers might limit this model for your patients?<br>What enablers would need to be in place for this to work?<br>General questions:<br>What proportion of your patients would this model work for?<br>Low–Mid–High<br>Can you think of anything about it that might be impractical?<br>11. Can you think of anything about it that might be unachievable? |

the perspective of delivery of care. Researchers prompted participants to elaborate where necessary. Detailed notes on the discussions were taken by the second researcher. Scenarios and schedule of questions were refined through a pilot focus group with people from outside of the geographic area, who then gave feedback.

**Analysis.** Handwritten notes from workshops were de-identified and aggregated into a consumer, and a provider narrative summary (in Microsoft Word) which followed the structure of the question framework. The narrative was then analysed deductively by two senior members of the research team (AC and NR) by collating and synthesising barriers and facilitators, risks, and benefits of the specialist model of care. Verbatim or paraphrased quotes from the handwritten notes were matched to each point. Results were verified by the whole team. Survey responses were analysed using the Microsoft Excel software tool to generate descriptive statistics of the participants' demographics. These data were compared to the catchment demographics to assess how well they represented the local community.

## Results

### Demographics

Twenty-two consumers from across the health catchment participated in five workshops, and two of these took part in a follow up interview. Forty-two health providers, from a diverse range of professional roles participated in five workshops. Data saturation was reached in both groups. Demographic data, compiled from the pre-workshop survey were aggregated to determine the representativeness of the sample (Table 2).

**Consumers.** For consumers' health conditions, there was a spread across the major physiological systems with the majority having experienced cardiac, lung or bone related conditions. This is representative of the catchment where cardiac, chronic pulmonary disease and musculoskeletal problems are listed in the five most common causes of hospitalisation (see S2 File) [19]. All the consumers were proficient in English with 23% speaking another language at home (e.g., Punjabi, Serbian). Although most of the consumers identified as Australian (82%), there was evidence of ethnic diversity (e.g., Indian, Middle Eastern), which also reflects the demographics of the catchment.

**Providers.** Fifty-five percent of the providers worked in the LHD while the remaining 45% worked in areas outside of the LHD, but within the new hospital catchment. The providers worked in a variety of professional roles including management, nursing, allied health, medical, general practice and administration. Allied health professionals included physiotherapists and speech pathologists (Fig 1)

**Table 2. Consumer and provider demographics.**

|  | Consumer (n = 22) | Provider (n = 42) |
|---|---|---|
| Gender |  |  |
| Male | 9 | 14 |
| Female | 13 | 28 |
| Age |  |  |
| Under 30 | 3 | 10 |
| 31 to 45 | 5 | 17 |
| 46 to 60 | 11 | 11 |
| Over 61 | 2 | 4 |
| Prefer not to say | 1 | 0 |

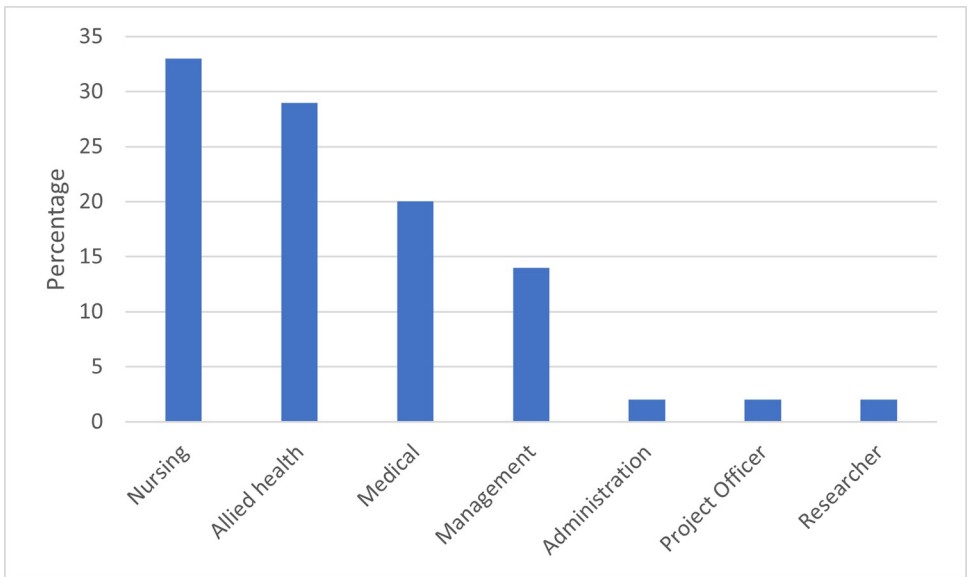

**Fig 1. Distribution of provider roles in focus groups.**

The providers reported having a diverse range of medical specialties with most having an 'other' speciality such as obstetrics and psychiatry (33%), then bone (16%), lung (15%), abdominal (12%), heart (10%), psychology (7%), and renal dialysis (8%).

Most of the providers rated their English proficiency as excellent (95%) with 31% speaking another language at home (e.g., Italian, Mandarin). Although most of the providers identified as Australian (83%), there was evidence of ethnic diversity (e.g., South American, Aboriginal and Torres Strait Islander).

### Workshops and interviews

**Summary of findings.** The specialist care unit model presented was viewed favourably by consumers and providers as being convenient, patient-focused and likely to deliver high quality care. It was also noted as being easy to navigate (in contrast to other models of care) as every service that was needed was there in the unit. Both consumers and providers valued the sense of reassurance that a specialist team brings. Providers were positive about the model, stating evidence of good outcomes from such specialist units and that they "rarely hear negative feedback" about the model. Some consumers were uncertain about what they perceived as a narrow focus of the specialist units and wondered if this would mean co-morbidities might be overlooked or not treated as well, as by a generalist. There was also debate among both consumers and providers as to whether specialist units would match the particular needs of the community. Key issues identified by the participants are shown in Table 3. We have structured the results below under the broad headings of: (i) barriers, challenges and risks (for consumers then providers), and (ii) enablers and opportunities. Extracts from the notes (paraphrased or shorthand quotes) are given to illustrate issues raised in the workshops.

**Barriers, challenges, and risks.** Consumers and healthcare providers identified barriers and challenges that may be encountered when delivering or receiving care in specialist units. Key themes identified by both consumers and providers included accessibility, a perceived narrow scope of practice, patient wellbeing, coordination with other services, resources and infrastructure, and awareness and expectations of the unit within the hospital and the community.

**Table 3. Themes from the workshops for consumer and provider participants.**

| Theme | Barriers, challenges and risks | | Enablers and opportunities | |
|---|---|---|---|---|
| | Consumers | Providers | Consumers | Providers |
| **Accessibility** | Not local<br>Poor public transport | Limited capacity of units | Local<br>Easy access built in | |
| **Narrow scope of practice** | Poor fit for some patients with multi-morbidities | Few patients may meet all the criteria;<br>Unable to deal with co-existing conditions | | |
| **Resource and infrastructure concerns** | Understaffing, inexperienced staff | Less flexible use of beds | Competent staff and stable funding | |
| **Patient wellbeing** | Unit may not suit all patients;<br>Alternative models of care may be better | | | |
| **Coordination with other services / departments** | | May be difficult to appropriately triage and place patients;<br>Need for complementary models of care | Access to complementary services built in | Streamlined admission and referral processes;<br>Collaboration across departments;<br>Flexibility in admission criteria |
| **Awareness and expectations** | | Uncertain if demand would be high or low | | Knowledge of the unit's benefits and limitations would foster appropriate use |
| **Expert care** | | | High quality care as providers were experts;<br>One stop shop as allied health co-located | Well trained, appropriately experienced staff |

*Consumers. (i) Accessibility.* The first issue that consumer participants identified was accessibility to the specialised unit. Specialist units were seen as few and far between and may involve burdensome travel. Issues around infrequent, inconvenient, or unsafe public transport, lack or cost of parking, and problems with wayfaring in an unfamiliar hospital were all mentioned. Travelling to a distant specialist unit when an acute complication arises may not be an option if there is a closer general facility. The distance may also be a barrier to family or friends who wish to support the patient. Specialist ambulatory units may not be available out of hours and may negatively influence the decision to seek care in a timely manner. There were also fears that specialist units may not be publicly funded, resulting in increased costs to patients.

*Complications may not be prioritised if they have to travel to the specialist unit.* [Consumer 1 interview 2]

*(ii) Narrow scope of practice.* The second theme was around patient factors that consumers perceived may make the specialist unit a poor fit or even unsafe for some patients. There was discussion around possible mismanagement of "border-line" cases who may not fully meet the criteria for a specialist unit, potential for missed care that the specialist unit may not often manage, and a perception of a narrow, inflexible, specialist focus. Co-morbidities may mean the focus should be on a different condition to that of the specialty (e.g., a patient with stable, well-managed dementia but unstable angina in a dementia unit). Linked to the perceived narrow scope of care was uncertainty that if the patient's condition changed, would they be able to stay in the unit or would they have to be relocated. Moving to another unit was seen as causing possible disruption to already initiated care and distress to the patient. Broader, generalist services were seen as perhaps more suitable and safer for patients with complex care needs. There was also uncertainty around the appropriateness of care in an inpatient specialist unit rather than other models such as 'hospital in the home'.

*Specialist units have a too narrow focus*. [Consumer 1 Workshop 6]

*How do the medical staff choose where to admit the patient if they don't fit the criteria for the special unit but still need that specialized service*? [Consumer 3 Workshop 8]

*Is it appropriate for this patient to be treated in this environment*? *Does he really need to be in the hospital for this or can it be treated at home*? [Consumer 1 Workshop 8]

*(iii) Patient wellbeing.* Patient wellbeing was also questioned in a specialist unit, in one case triggered by a prior negative experience of such a unit. In contrast to the hospital in the home model, consumers thought that patients may be anxious or overwhelmed by the unfamiliar setting (especially if they are living with dementia) and the large spectrum of patient conditions in the unit could be detrimental. There was also a perception among some participants that patients would be cut off from home, family or other supports if in a specialist unit. Some consumers called for adjacent family or carer accommodation to facilitate support for the patient.

*The patient is not in their own environment—it may be a trigger for anxiety*. [Consumer 2 Workshop 8]

*(iv) Resource and infrastructure concerns.* Consumers identified resource and infrastructure concerns associated with the specialist unit model. Limited funding could lead to understaffing or use of untrained or inexperienced staff which in turn would lead to a poorer quality of care. It could also result in equipment or other resources being suboptimal. Limits to specialist unit capacity was seen as another risk which required general services as a back-up. Investing in specialist units may not be appropriate for the needs of the community and may limit services.

*The specialist unit is only good if the general hospital can handle overflow*. [Consumer 2 workshop 6]

*Providers*. Providers identified several barriers, difficulties and risks associated with specialist hospital care, covering many of the same issues as consumers. Providers discussed potential access barriers to specialist unit care including burdensome patient travel over long distances.
*(i) Accessibility.* Further barriers to access were the limited capacity of the units. Once full, there was a perception that there was nowhere else suitable to transfer patients. Specialist units are often well staffed with allied health professionals. If a patient cannot access a specialist unit bed, they are likely to be placed elsewhere in the hospital with limited allied health support which could be detrimental.

*The specialist unit could lack flexibility or flex beds to put overflow patients elsewhere in the hospital*. [Provider 2 Workshop 6]

*Sometimes these models become so popular they are victims of their own success*. [Provider 1 Workshop 4]

*(ii) Narrow scope of care.* The narrow scope of practice in a specialist unit was a key concern. Providers discussed the "tunnel vision" that can result from a too narrow focus of care and the trend toward more specialisations when what was needed were more mixed specialties. They also speculated that if criteria are too tight, then maybe no one fits, making it a redundant

service. Further, inflexible inclusion/exclusion criteria could lead to a lack of choice for patients and their families about where they are best treated.

*The Unit [develops] tunnel vision and end up having no access to dealing with other conditions.* [Provider 1 Workshop 8]

*Need the ability to access all hospital resources when you want to access them. Too much specialisation means you lose the ability to care for simple things.* [Provider 5 Workshop 2]

Safety issues were raised by providers. Co-morbidities were frequently mentioned with a perception that a specialist unit may not be able to successfully diagnose and manage other conditions. Concurrent mental illness was cited as an example of where placement in a specialist medical unit may be inappropriate. In addition, there was a perception that the system lacked robust referral protocols that ensured patients could access the care they needed from other health providers.

*Depends on the setting facility that the patient needs. For example, if a patient with mental health issues is in a cardiac-specific unit, the patient may decompensate their mental health while in care of cardiac unit.* [Provider 4 Workshop 2]

**(iii) Resources and infrastructure concerns.** Providers identified potential resource barriers and health system factors that may affect specialist hospital care. Limited capacity to take other patients may lead to bed block, or less flexible bed management if there is a regional crisis and a need for extra beds. Design of the wards and the specialisation of the staff may make them a poor fit for other patients. Providers speculated that specialist hospitals and units may not be integrated well with other hospitals and services in the district.

**(iv) Coordination with other services or departments.** Coordination and quality of care were also identified as barriers. Admission barriers included difficulties in appropriately placing patients, and administration and triage concerns.

*We'd face frustration every day in terms of which specialty the patients should fall under. One solution is to use admission matrices that help decision making regarding who [which specialty team] patients get admitted under.* [Provider 1 Workshop 2]

*There's a risk of hot potato-ing in terms of what is the underlying condition?* [Provider 1 Workshop 2]

Also, in line with consumers, there was speculation around whether a specialist unit would suit the needs of the area as they only serve a subset of patients. Providers noted that smaller hospitals would struggle to adequately staff and resource a specialist unit which were perceived as expensive to establish and operate.

**(v) Awareness and expectations.** There were conflicting views on patient demand for specialist units. Some thought there would be a lack of patient interest and so low demand, while others believed they could be overwhelmed.

*Everyone goes there because they know it is good, then the number of beds becomes an issue.* [Provider 1 Workshop 6]

**Enablers and opportunities.** Across workshops, consumers and providers identified benefits of the model, and supporting factors or enablers that may assist in delivering high quality,

specialist unit care. Key themes identified by both consumers and providers included: systems and processes, accessibility, people, comfort and environment, resources and infrastructure, education and skills.

*Consumers. (i) Expert care.* The benefits of the model were clearly articulated by the consumers. The high quality of care and the ease of a "one stop facility" were mentioned frequently. Specialist trained staff were seen as more compassionate and knowledgeable about the condition and so more likely to give high quality care. As a result, the patient's outcomes would be better. The idea of wrapping appropriate, specialised services around the patient was noted several times as particularly valuable to shift the focus onto the patient:

> *The focus of care is on the patient and not on the provision of services.* [Consumer 1 interview 2]

> *It's important to have all the services in the one unit that are specialised.* [Consumer 1 workshop 2]

*(ii) Coordination with other services or departments.* Consumers described the importance of systems and processes to support the delivery of specialist hospital care. Participants spoke of needing efficient access to other services, including sharing of medical records, mitigating the perceived hazard of missing care for co-morbidities.

> *You would need access to records from different facilities, in case the patient needs other special services.* [Consumer 4 Workshop 6].

*(iii) Accessibility.* Physical accessibility was raised as an important enabling factor for specialist hospital care, such as drop off and pick up spots, proximity to public transport, and services for culturally and linguistically diverse patients. Appropriate security to maintain the safety of patients, staff and visitors was also mentioned for high-risk units.

> *It would be easy for family members to transport the patient to the unit if site is local.* [Consumer 3 workshop 6]

*(iv) Competent staff and stable funding.* Finally, consumers discussed at length the importance of able and competent multidisciplinary staff and stable funding.

> *Staff need adequate training and education for care and appropriate assessment, and to designate the patient to the correct specialized care.* [Consumer 3 Workshop 8].

*Providers. (i) Awareness and expectations.* Providers identified patient, staff, and community education as key enablers for the success of the proposed specialist unit model. This included ensuring the community and potential referrers (e.g., GPs) are aware of the benefits and limitations of the model, and do not have unrealistic expectations.

> *If somebody turns up with something even as simple as a urinary tract infection–everybody needs to know the plan in the event of a deterioration. This does not mean just listening to the family.* [Provider 3 Workshop 4]

*(ii) Coordination with other services or departments.* Providers identified key systems and processes as enablers for the specialist unit model including streamlined admission pathways such as rapid assessment protocols for patients that do not need to go through Emergency. A high degree of collaboration across departments was seen as necessary to support these

pathways, and to transfer care easily if needed. This relied too on a good and flexible culture among all the hospital's staff, and some flexibility in the admission criteria.

> *You can't know everything, even in a specialised unit [so one needs to collaborate with other health professionals].* [Provider 1 Workshop 4]

> *Implementing processes and pathways in specialised units means that all the staff know what needs to be done for that patient at different stages in their care and the expectations for discharge.* [Provider 1, workshop 4]

> *Integration with multidisciplinary care is needed. For example, [Hospital X] had a mental health unit connected to the general hospital and flow of patients between units was easy.* [Provider 1 Workshop 2]

> *You need some leeway in the criteria so people get in when they need to, but not too loose that everyone ends up there.* [Provider 1 Workshop 6]

As in the consumer workshops, providers emphasised the key role of well-trained appropriate staff including specialist staff, multidisciplinary teams and specialist emergency teams. Providers also commented on the need for complementary models of care, especially integrated care. This would ensure referrals and transfers to other health services outside of the specialist unit were smooth.

## Discussion

Perceptions of the specialist unit model of care were explored in five consumer and five provider workshops, involving participants from a single local health district catchment planning a new facility. The overall sentiment was positive with frequent references to high quality, expert care, and specialised staff including allied health being provided by the specialist unit. This is closely aligned with the many reports in the literature of superior clinical outcomes and patient satisfaction from specialist units [e.g., 9–11]. Greater knowledge and experience with management of a single primary diagnosis intuitively suggests better care and outcomes, and greater confidence from consumers [20]. The same issues that were named as barriers or risks to the model operating efficiently, were also often named as enablers; their benefit or risk linked closely to their presence or absence: systems and processes, accessibility, staff qualities, a patient-centred approach, resources and infrastructure, and managing expectations.

The "tunnel vision" and "narrow focus" of the specialist unit was mentioned in both consumer and provider workshops as a drawback of the model. This aligns with a studies that found generalist clinicians with a broad knowledge of how to manage patients with multi-morbidities rather than a single condition were often seen as preferred doctors e.g., [21]. There was an assumption, particularly among some consumers, that the primary condition that the unit specialised in, resulted in a very narrow and limited scope of practice, which may have quality and safety issues for the patient. While there is little evidence from the literature to support this, Kreindler and colleagues [22] take a systems level view and state that specialist services can be inflexible and standardised in their processes, suiting only a narrow cohort of patients.

Experienced and well-educated staff were agreed by all participants to be necessary to run these units well. Criteria for admission to the unit was discussed at length and it was agreed that they should be flexible and focussed on the best outcome for the patient. As more and more patients are admitted with multiple or complex conditions, this will be important to ensure patients are directed to the care most appropriate to them.

Specialist units are embedded in a larger health system and so cannot be considered to operate independently. It was emphasised by both groups that beneficial as a specialist unit may be, flexibility and integration with the larger health system–other specialist or generalist wards, community services such as palliative care or rehabilitation—were essential to efficiently manage complications or changes in patients' needs. The limits of specialist clinicians' knowledge and expertise, flagged as a barrier, could be mitigated by collaboration and shared care across the whole organisation. Consumers implied, and providers explicitly stated that this combination of the specialist unit and integrated care was the ideal model. The importance of robust referral pathways and structured, as well as informal networks all support this blended model.

Both types of participants spoke about patient centred approaches which included family and emphasised welcome and comfortable surroundings. The scenario provided was a dementia unit where the presence of familiar people and "homely" surroundings is important to reduce confusion and distress. If the specialist unit example had been a cardiac unit or high-risk foot service for people with diabetes, this aspect may not have been so prominent in participants' minds. However, many specialist services are multidisciplinary including nurses, physiotherapists, dietitians or psychologists. These professions tend to have a more holistic and person-centred paradigm reinforcing the perception of this as a benefit of the model [23].

Some issues that were raised were widely applicable to the health system not just the specialist unit model. Physical access to health services generally is a perennial problem with frequent complaints about scarce or expensive parking, poor public transport services and inadequate wayfaring once inside the facility. The fact that these issues continue to be flagged by consumers as the most pressing issue they face to access health care show these issues have not yet been given adequate attention by policymakers and infrastructure designers.

Specific to the specialist model of care was the perception that they were "few and far between", with a limited number of specialised units for any condition. This may require consumers to travel outside their local area to a single centralised service. Having a local specialist unit that caters for one's condition was seen as a valuable asset (due to less burden to access), but if one's condition required traveling to a distant unit this was a liability. Patients from regional and remote areas may need to travel to their local facility if a specialist unit was established there. This prompted comments around the need for family accommodation to alleviate the burden of travel. Interestingly, consumers did not discuss access to the units themselves, taking for granted that if the patient met the criteria they would be admitted. This is in contrast to an increasing body of literature showing difficulties around access to specialist services leading to unmet needs and poorer clinical outcomes (e.g., [24, 25]).

The perceptions of our consumer and provider participants were considered important to factor into the planning of a new community health facility and were based on their experience of existing services. The design of appropriate services, of course should also be firmly based on an analysis of the community's health needs to avoid "population–capacity misalignment", a situation where the services provided do not meet the community's needs [22]. Kreindler and colleagues [22] call for detailed examination of the health needs of the community before deciding on services: for example, high demand for surgery may be best served by specialist, standardised surgical services, while high demand for services for frail elderly patients with complex care needs would indicate more broad and flexible, generalist services. The current study did not include a detailed analysis of local demand for services but comments around the desirability of local specialist services and the need for coordination with integrated care models to support flexibility, align with this concept. Both consumers and providers discussed the importance of choosing a specialist service that was of most relevance to the community.

## Recommendations for specialist units

These findings prompt some high level recommendations for the implementation of specialist units that avoid some of the perceived drawbacks. First, specialist units should be determined by a condition's prevalence in the community and placed in areas of most need. The benefit of a local service can then be realised for more people. Attention to public transport access, parking and, if required family accommodation should be included. Second, funding and infrastructure for the unit needs to be stable to avoid staffing shortages or inappropriate skills mix. Third, robust yet flexible admission and referral processes should be implemented with wide dissemination of benefits and limitations of these units. Fourth, units should be supported and complemented by other models of care such as hospital in the home or integrated care to ensure smooth transitions into and out of the unit.

## Strengths and limitations

This study took advantage of a unique opportunity to explore the perceptions of consumers and providers on a range of models of care to inform the design of a proposed new local health facility. Strengths of the study included keen interest and engagement of local residents to take part (limited only by the researchers' capacity to hold workshops) and the representative sample of this diverse population. The focus on conditions that were known to have a high local prevalence was also a strength making the research engaging and relevant to participants.

Limitations of the work included possible bias of the participants who self-selected and were motivated to have a say in the health facility design. Their views may not necessarily be representative of the rest of the local community. The choice of a dementia specialist unit as the example under discussion may also have limited participants' responses. The distinctly local and tailored nature of the study means that generalisability should be exercised with caution.

## Conclusions

The specialist unit model of care was perceived as providing holistic and high quality care and where it was supported by robust processes and systems and well-trained staff, was a desirable element to include in the design of the proposed new local health facility. Drawbacks included unrealistic expectations of the public (both overly pessimistic or optimistic), potential for unclear admission criteria or a too narrow scope of practice. Providers stated (and consumers implied) that integration of the service with other parts of the health system were essential so that patients with changing needs could be effectively transferred to more appropriate care settings. Generic problems for consumers with accessing care such as poor public transport service and inadequate parking were also mentioned. More relevant to the specialist model was the perception of these units being "few and far between", meaning a significant drawback may be the need to travel long distances to access one.

## Supporting information

**S1 File. COREQ (COnsolidated criteria for REporting Qualitative research) checklist.**
(PDF)

**S2 File.**
(PDF)

## Acknowledgments

We thank our consumer and health provider participants and our partners at Health Infrastructure.

## Author Contributions

**Conceptualization:** Ann Carrigan, Natalie Roberts, Robyn Clay-Williams, Peter D. Hibbert, Katherine Maka, Graeme Loy, Jeffrey Braithwaite.

**Data curation:** Natalie Roberts.

**Formal analysis:** Janet C. Long, Ann Carrigan, Natalie Roberts, Robyn Clay-Williams, Peter D. Hibbert.

**Funding acquisition:** Ann Carrigan, Robyn Clay-Williams, Peter D. Hibbert, Katherine Maka, Graeme Loy, Jeffrey Braithwaite.

**Methodology:** Janet C. Long, Ann Carrigan, Robyn Clay-Williams, Yvonne Zurynski, Jeffrey Braithwaite.

**Project administration:** Ann Carrigan, Natalie Roberts.

**Supervision:** Ann Carrigan, Natalie Roberts, Robyn Clay-Williams, Peter D. Hibbert.

**Validation:** Janet C. Long, Ann Carrigan, Peter D. Hibbert, Yvonne Zurynski, Katherine Maka, Graeme Loy, Jeffrey Braithwaite.

**Writing – original draft:** Janet C. Long.

**Writing – review & editing:** Janet C. Long, Ann Carrigan, Natalie Roberts, Robyn Clay-Williams, Peter D. Hibbert, Yvonne Zurynski, Katherine Maka, Graeme Loy, Jeffrey Braithwaite.

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
