## [Decision Letter · Decision Letter 0]

9 Aug 2023

PONE-D-23-12601Consumer and provider perceptions of the specialist unit model of care: a qualitative study.PLOS ONE

Dear Dr. Long,

Thank you for submitting your manuscript to PLOS ONE. After careful consideration, we feel that it has merit but does not fully meet PLOS ONE’s publication criteria as it currently stands. Therefore, we invite you to submit a revised version of the manuscript that addresses the points raised during the review process. Please ensure that your decision is justified on PLOS ONE’s publication criteria and not, for example, on novelty or perceived impact.

We look forward to receiving your revised manuscript.

Kind regards,

Caroline Watts, PhD

Academic Editor

PLOS ONE

Journal Requirements:

4. Please amend the manuscript submission data (via Edit Submission) to include author  Ann Carrigan.

Reviewers' comments:

Reviewer's Responses to Questions

**Comments to the Author**

1. Is the manuscript technically sound, and do the data support the conclusions?

Reviewer #1: Partly

Reviewer #2: Yes

2. Has the statistical analysis been performed appropriately and rigorously? 

Reviewer #1: N/A

Reviewer #2: N/A

3. Have the authors made all data underlying the findings in their manuscript fully available?

Reviewer #1: Yes

Reviewer #2: Yes

4. Is the manuscript presented in an intelligible fashion and written in standard English?

Reviewer #1: Yes

Reviewer #2: Yes

5. Review Comments to the Author

Reviewer #1: Thank you for the opportunity to review this manuscript. I enjoyed reading the article and especially appreciated the immediate relevance of the research to informing health system design.

The article aims to explore the views of health care consumers and providers on the benefits and challenges surrounding a specialist unit model of care. The research is a qualitative study, drawing on data obtained from workshops and interviews with consumers and providers in a metropolitan area of Australia. The authors identify a number of potential challenges and benefits arising from the specialist unit model of care. The research outlined in the article provides valuable information for policymakers seeking to develop innovative health care models in NSW and elsewhere.

As outlined above the research addresses important issues in designing new models of care delivery. However, from my perspective there are some flaws with the article. I have outlined my major concerns in the attached document, as well as identifying a number of minor issues.

Reviewer #2: This is a clearly written paper that describes outcomes of a broad consultation process, discussing potential models of care for a new health centre. It is encouraging to see this type of work being done in the development phase of new health facilities, and the paper will make an important contribution to the literature.

There are two aspects of the paper that I think could be improved prior to publication.

Firstly, the context session providing information about the broader consultation and place of this study within it is clear and helpful. I suggest that it would be best to keep all the information about the broader study here, and then have the rest of the method focus only on the current consultation about specialist care models to avoid any confusion. Hence delete the first sentence under workshops, and instead add a sentence in the context section to state that the consultation process involved a series of workshops (perhaps include the total number of workshops and sample size to give an indication of the overall scope?) with each workshop focussing on a different model of care. Prior to the last sentence of paragraph 1 under context would be a good a place for this (about line 85).

Secondly, the discussion is largely a further presentation of results. It provides the reader with more detail about what the participants said, and summarises the data in different ways, but there is limited integration of the data with what else is known about this field. For example:

- Paragraph 2 talks about “tunnel vision” but includes no references. Are there any studies that have discussed this issue that can be integrated here? The authors state that consumers were making an assumption – is there any data that challenges or supports this assumption from other studies?

- Paragraph 3 talks about integration, and states that “Consumers implied, and providers explicitly stated that this combination of the specialist unit and integrated care was the ideal model”. What do others think about whether this is the ideal model? Does the literature support the view of participants in your study? Sara Kreindler has done some interesting work on this based on a qualitative study involving more than 300 interviews with health services managers in Ontario. She talks about how specialist vs generalist models can be best used, depending on population need (PMID: 33949820). It would be interesting to consider how her findings align (or not) with the perceptions of this group.

- The final paragraphs of the discussion focus on access to care, presenting participants’ perceptions but not references to other literature. There is a lot of literature on access (or lack of access) to specialist care that could be used as a basis for a broader discussion of this issue raised.

- I note that one issue not discussed in the paper is how one would choose the specialties to be catered for in a specialty model. Did this come up at all in the workshops? If not, was that because the issue was not raised in response to the prompts provided? It seems to me that specialist are services most likely work well for those they cater for but what are the implications for those who don’t meet the criteria? There are far more potential areas of specialty than could ever be addressed with a limited number of specialty units in a new health service. This is a potentially complex issue that would surely need to be a key consideration if a new health service was to decide to focus on specialty care models.

6. PLOS authors have the option to publish the peer review history of their article (what does this mean?). If published, this will include your full peer review and any attached files.

Reviewer #1: No

Reviewer #2: No

---

## [Author Response · Author response to Decision Letter 0]

26 Sep 2023

We have uploaded an attachment "PONE-D-23-12601 Author response to reviewers" containing an itemised response to all the comments provided by the reviewers.

---

## [Editor Report · Decision Letter 1]

4 Oct 2023

Consumer and provider perceptions of the specialist unit model of care: a qualitative study.

PONE-D-23-12601R1

Dear Dr. Long,

We’re pleased to inform you that your manuscript has been judged scientifically suitable for publication and will be formally accepted for publication once it meets all outstanding technical requirements.

Kind regards,

Caroline  Watts, PhD

Academic Editor

PLOS ONE

---

## [Editor Report · Acceptance letter]

12 Oct 2023

PONE-D-23-12601R1 

Consumer and provider perceptions of the specialist unit model of care: a qualitative study. 

Dear Dr. Long:

I'm pleased to inform you that your manuscript has been deemed suitable for publication in PLOS ONE. Congratulations! Your manuscript is now with our production department. 

Kind regards, 

on behalf of

Dr. Caroline Gay Watts 

Academic Editor

PLOS ONE